# A stepped-wedge randomised trial on the impact of early ART initiation on HIV-patients' economic outcomes in Eswatini

Janina Isabel Steinert[1]*, Shaukat Khan[2], Khudzie Mlambo[2], Fiona J Walsh[2], Emma Mafara[2], Charlotte Lejeune[2], Cebele Wong[2], Anita Hettema[2], Osondu Ogbuoji[3], Sebastian Vollmer[4], Jan-Walter De Neve[5], Sikhathele Mazibuko[6], Velephi Okello[6], Till Bärnighausen[5], Pascal Geldsetzer[5,7]

[1]Technical University of Munich, Munich, Germany; [2]Clinton Health Acccess Initiative, Boston, United States; [3]Center for Policy Impact in Global Health, Duke Global Health Institute, Duke University, Durham, United States; [4]University of Goettingen, Goettingen, Germany; [5]Heidelberg Institute of Global Health, Heidelberg University, Heidelberg, Germany; [6]Ministry of Health of the Kingdom of Eswatini, Mbabane, Eswatini; [7]Division of Primary Care and Population Health, Department of Medicine, Stanford University, Stanford, United States

*For correspondence: janina.steinert@tum.de

Competing interests: The authors declare that no competing interests exist.

## Abstract

**Background:** Since 2015, the World Health Organisation (WHO) recommends immediate initiation of antiretroviral therapy (ART) for all HIV-positive patients. Epidemiological evidence points to important health benefits of immediate ART initiation; however, the policy's impact on the economic aspects of patients' lives remains unknown.

**Methods:** We conducted a stepped-wedge cluster-randomised controlled trial in Eswatini to determine the causal impact of immediate ART initiation on patients' individual- and household-level economic outcomes. Fourteen healthcare facilities were non-randomly matched into pairs and then randomly allocated to transition from the standard of care (ART eligibility at CD4 counts of <350 cells/mm3 until September 2016 and <500 cells/mm3 thereafter) to the 'Early Initiation of ART for All' (EAAA) intervention at one of seven timepoints. Patients, healthcare personnel, and outcome assessors remained unblinded. Data were collected via standardised paper-based surveys with HIV-positive adults who were neither pregnant nor breastfeeding. Outcomes were patients' time use, employment status, household expenditures, and household living standards.

**Results:** A total sample of 3019 participants were interviewed over the duration of the study. The mean number of participants approached at each facility per time step varied from 4 to 112 participants. Using mixed-effects negative binomial regressions accounting for time trends and clustering at the level of the healthcare facility, we found no significant difference between study arms for any economic outcome. Specifically, the EAAA intervention had no significant effect on non-resting time use (RR = 1.00 [CI: 0.96, 1.05, p=0.93]) or income-generating time use (RR = 0.94, [CI: 0.73,1.20, p=0.61]). Employment and household expenditures decreased slightly but not significantly in the EAAA group, with risk ratios of 0.93 [CI: 0.82, 1.04, p=0.21] and 0.92 [CI: 0.79, 1.06, p=0.26], respectively. We also found no significant treatment effect on households' asset ownership and living standards (RR = 0.96, [CI 0.92, 1.00, p=0.253]). Lastly, there was no evidence of heterogeneity in effect estimates by patients' sex, age, education, timing of HIV diagnosis and ART initiation.

**Conclusions:** Our findings do not provide evidence that should discourage further investments into scaling up immediate ART for all HIV patients.

**Funding:** Funded by the Dutch Postcode Lottery in the Netherlands, Alexander von Humboldt-Stiftung (Humboldt-Stiftung), the Embassy of the Kingdom of the Netherlands in South Africa/

Mozambique, British Columbia Centre of Excellence in Canada, Doctors Without Borders (MSF USA), National Center for Advancing Translational Sciences of the National Institutes of Health and Joachim Herz Foundation.
**Clinical trial number:** NCT02909218 and NCT03789448.

## Introduction

Recent trials have pointed to substantial health benefits of immediate antiretroviral therapy (ART) initiation for all HIV-positive patients compared to initiating ART based on a CD4-cell count threshold. Benefits include reduced HIV-related mortality and morbidity and decreased transmission risk to HIV-negative sexual partners (*Danel et al., 2015*; *Cohen et al., 2016*; *Lundgren et al., 2015*; *Hayes et al., 2019*; *Ford et al., 2018*). In line with this epidemiological evidence, the World Health Organization (WHO) has updated its consolidated guidelines on the use of antiretrovirals in 2015, now advocating for immediate ART initiation (or 'universal test and treat') for all HIV-positive adults, adolescents, and children (*WHO, 2019*).

In view of these major changes in ART provision, it is crucial for health policy makers to understand the implications of immediate ART initiation for HIV patients' economic outcomes. At high CD4-count levels, we would expect the majority of patients to be relatively healthy, and thus have productivity levels and out-of-pocket health expenditures that are similar to those of HIV-negative patients (*Thirumurthy et al., 2013*). While ART may still improve economic welfare among these patients through an improvement in health status, it may also decrease these patients' economic welfare through, for example, the side effects of antiretroviral drugs, increased frequency of (ART) clinic visits or stigma from taking ART (*daCosta DiBonaventura et al., 2012*; *Unge et al., 2008*). The economic consequences of early ART initiation for this specific patient group are therefore unclear and have to date not been investigated experimentally.

Previous studies have assessed labour market outcomes and overall financial wellbeing of patients on ART, relative to patients not yet on ART. Of these, several studies have highlighted beneficial economic impacts of ART initiation, which are primarily based on the positive labour market effects of improved health. Accordingly, empirical evidence has pointed to higher work performance and productivity, (*Bor et al., 2012*; *Larson et al., 2008*; *Beard et al., 2009*; *Thirumurthy et al., 2008*) lower absenteeism, (*Larson et al., 2008*) increases in savings rates, (*Baranov and Kohler, 2018*) as well as increased educational expenditures and attainment (*Baranov and Kohler, 2018*; *Lucas et al., 2019*) following ART initiation. Conversely, other studies have documented detrimental economic effects of ART initiation (even under universal access to ART schemes), largely driven by three suggested mechanisms: first, by increased patient-borne healthcare expenditures associated with travel to ART clinics, clinic and hospital fees, and income foregone; (*Rosen et al., 2007*; *Chimbindi et al., 2015*; *Leive and Xu, 2008*) second, by elevated levels of food insecurity due to a treatment-induced increase in appetite and fewer financial resources to absorb the higher food expenditures; (*Patenaude et al., 2018*) and third, by reduced productivity as a result of short-term adverse and toxic effects linked to antiretroviral drugs (*Danel et al., 2015*; *Rosen et al., 2007*). However, these previous studies provide only little insights on the anticipated economic effects of immediate ART initiation because they are based on outdated treatment guidelines, thus comparing HIV-patients above and below a certain CD4 cell count level (e.g. 500 cells/mm$^3$) that determines ART eligibility. Given that HIV-patients who are not yet on ART but have a relatively low CD4 count may be more susceptible to opportunistic infections and adverse events than those with higher CD4 counts, this comparison group is inadequate for assessing the economic consequences of the current WHO-endorsed ART initiation strategy that is independent of patients' CD4 counts.

To decide whether and how much governments and international organisations should invest in scaling up immediate ART initiation for all HIV-patients, it is crucial to understand the impact of immediate ART initiation not only on health but also on HIV patients' economic outcomes. This is the first randomised trial aimed at answering this question. Specifically, we conducted a stepped-wedge cluster-randomised controlled trial of the 'Early Initiation of ART for All' (EAAA) intervention for HIV-patients in Eswatini to test the causal impact of immediate ART initiation on a range of economic outcomes, including patients' time use, employment, household expenditures, and household living standards.

**eLife digest** Human immunodeficiency virus (HIV) is an incurable virus that attacks the immune system and affects around 39 million people worldwide. Once diagnosed, HIV can be treated with antiretroviral therapy (ART) to limit its effects and stop it spreading to other people. HIV rates vary across the world, but the African country of Eswatini has the highest prevalence with more than one in four (27%) people classed as HIV-positive.

Until 2015, people living with HIV were typically only treated with ART once their immune system weakened. Recent studies found that starting treatment earlier enhances the positive effects of ART. This caused the World Health Organization (WHO) to change their guidelines and advise people living with HIV to begin ART as soon as they are diagnosed. While antiretroviral drugs are usually provided to patients free of charge, accessing care can be expensive for patients because of high transport costs or lost time from income-generating activities. This means starting treatment earlier and, thus, having more frequent healthcare visits, may result in a greater cost to the patient. The economic impact of this change is unclear, and for patients living in poverty, these added costs can affect their decision on whether to continue treatment.

Steinert et al. interviewed 3,019 HIV-patients from 14 health facilities in Eswatini who began treatment with ART either immediately after diagnosis or after their immune system became suppressed. Patients were asked about their time spent being active to generate income, employment status, monthly household expenditures, and household living standards. On average, beginning ART earlier appears to have had no large negative effects on the economic wellbeing of patients. The same results were found for patient groups defined by sex, education, age, and time spent taking ART.

These findings suggest that starting ART for HIV as soon as possible offers medical benefits and seems to have no large economic consequences for patients in the short term, even for poorer communities. This adds weight to the WHO advice on HIV treatment and supports the need to continue to deliver effective treatments to countries like Eswatini that have a high rate of HIV infection.

## Results

### Sample characteristics

Fourteen healthcare facilities ('clusters') were consecutively enrolled into the Maximising ART for Better Health and Zero New HIV Infections (MaxART) stepped-wedge trial and 3019 participants were interviewed over the duration of the study. The mean number of participants approached at each facility and time step varied from 3.5 to 112 participants (see *Figure 1*).

*Table 1* summarises sociodemographic information separately for two study samples. The full sample was composed of 3019 participants, sampled across 14 healthcare facilities. Participants enrolled into the EAAA intervention arm were on average aged 38.3 years (range: 18–85 years), 71.0% were female, 53.5% were married, and 56.0% had completed at least some secondary schooling. Participants in the standard of care group had similar characteristics: 74.3% were female, 56.6% married, and 56.0% had completed at least some secondary education.

The random subset of participants with household-level data on household expenditures and living standards was composed of 1485 patients who were also sampled across all 14 healthcare facilities. Overall, sociodemographic characteristics were very similar to those of the full sample.

### EAAA intervention impact on patient's economic outcomes

Time use

The intervention impacts on patient-level and household-level economic outcomes are presented in *Figure 2*. Histograms for all continuous outcome variables are presented in *Figure 2—figure supplements 1–4*. Participants in the EAAA group and in the standard of care group reported very similar levels of non-resting and income-generating time use. Non-resting time was approximately nine out of 24 hr in both study arms and the treatment effect was effectively null with an average marginal difference of only 0.6 min between groups (RR = 1.00, 95% CI: 0.96, 1.05, p=0.93). The

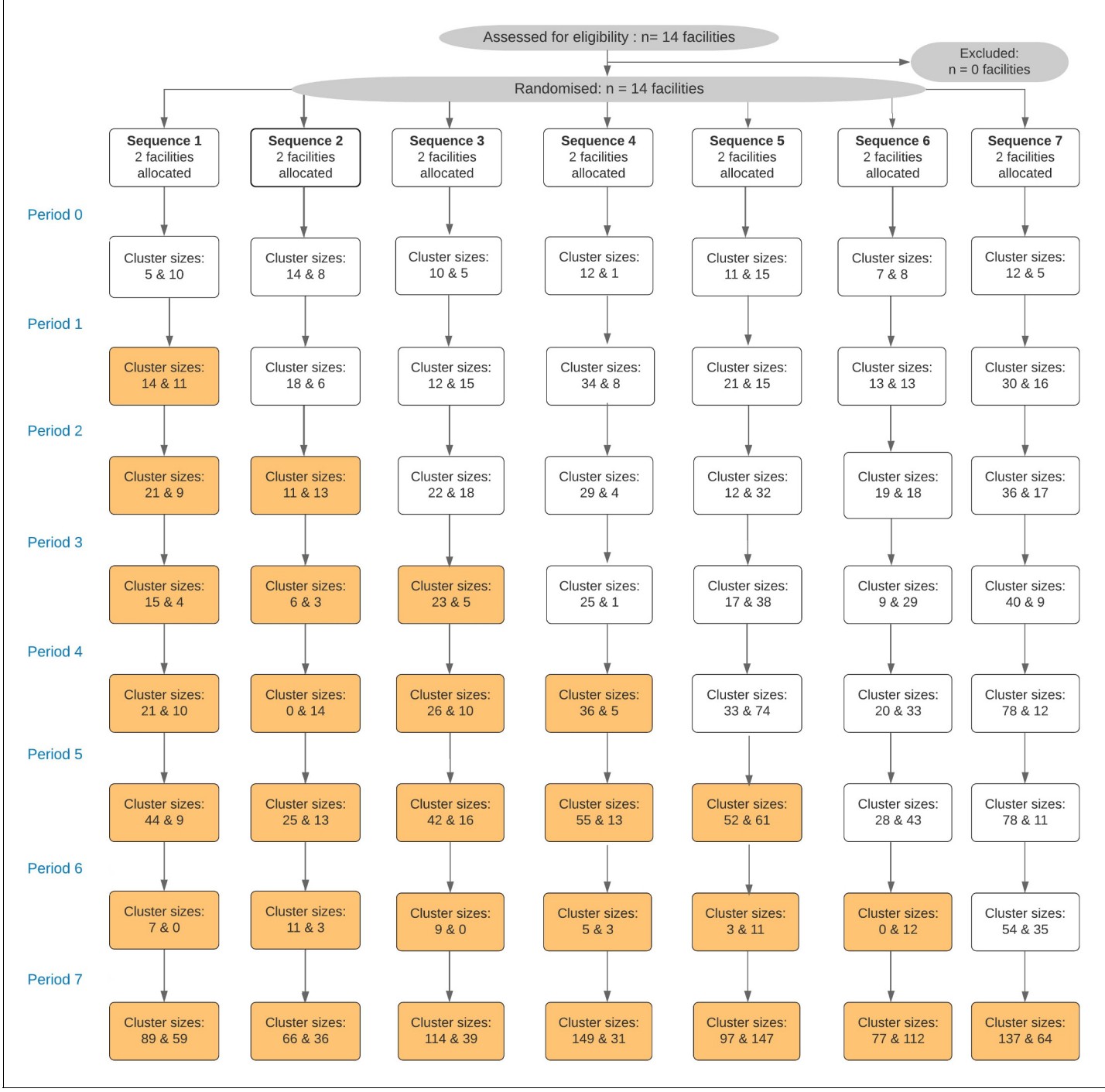

**Figure 1.** Participant flow chart (full sample).

treatment effect also remained precise and close to zero in alternative regression specifications, which included a random slope for time (see *Supplementary file 1A*). In addition, the results were similar when using a linear regression specification (β = 0.02, 95% CI: −0.36, 0.39, p=0.93, see *Supplementary file 1G*). Income-generating time was also similar between both groups (RR = 0.94, 95% CI: 0.73,1.20, p=0.61), translating into an average marginal difference of only 12.6 min between patients in the EAAA phase and patients in the standard of care phase. The difference was not statistically significant and remained similarly small in alternative specifications (see *Supplementary file 1B*).

**Table 1.** Sample characteristics.
**Full study sample (N = 3019)**

| | EAAA (N = 1868) | SoC (N = 1151) |
|---|---|---|
| Female, n (%) | 1326 (71.0%) | 855 (74.3%) |
| Age, mean (SD) | 38.3 (11.8) | 38.3 (11.8) |
| Education, n (%) | | |
| No formal schooling | 356 (19.1%) | 212 (18.6%) |
| Any primary schooling | 400 (21.4%) | 294 (25.5%) |
| Any secondary schooling | 1112 (59.5%) | 645 (56.0%) |
| Married, n (%) | 1000 (53.5%) | 651 (56.6%) |

**Random subsample with data on household expenditure and living standards (N = 1485)**

| | EAAA (N = 930) | SoC (N = 555) |
|---|---|---|
| Female, n (%) | 665 (71.5%) | 417 (75.1%) |
| Age, mean (SD) | 38.4 (11.9) | 38.2 (12.1) |
| Education, n (%) | | |
| No formal schooling | 175 (18.9%) | 99 (17.9%) |
| Any primary schooling | 192 (20.7%) | 142 (25.6%) |
| Any secondary schooling | 563 (60.5%) | 314 (56.6%) |
| Married, n (%) | 505 (54.3%) | 316 (56.9%) |
| Number of household members < 15 years, mean (SD) | 2.44 (1.11) | 2.58 (2.00) |
| Number of household members 15–60 years, mean (SD) | 2.75 (2.24) | 3.21 (2.22) |
| Number of household members > 60 years, mean (SD) | 0.35 (0.61) | 0.45 (0.78) |

Notes: Abbreviations: EAAA, SD = standard deviation.

## Employment

We observed a decline in general employment rates over the entire study period, from 0.64 (SD = 0.48) in study period 0 to 0.35 (SD = 0.48) in study period 7 (see *Figure 3*). The employment trend observed in our study population stands in contrast to the national employment rate during the same period, which remained constant at 77–78%. The difference in employment status between study groups was small and statistically insignificant (RR = 0.93, 95% CI: 0.82, 1.04, p=0.21, *Figure 2*). This finding remained robust across all alternative regression specifications (see *Supplementary file 1C*).

## EAAA intervention impact on household-level economic outcomes

### Household expenditures

Patients' total past-month household expenditures were 10% lower in the EAAA intervention group but this difference was not statistically different from zero (RR = 0.92, 95% CI: 0.79, 1.06, p=0.26, see *Figure 3*). All expenses were reported in Lilangeni (SZL) and converted into US dollars adjusted for purchasing power parity (PPP) and inflation for reporting purposes. This corresponds to a reduction in the mean expected monthly expenses of 105.83 SZL (95% CI: −289.042 to 77.38498 SZL), or 20.47 PPP\$ (95% CI: −55.93 \$ to 14.97 \$). Results remained virtually unchanged in alternative regression models (see *Supplementary file 1D*) or when imputing missing data using MICE (N = 1475) (see *Supplementary file 1E*).

### Household living standards

Lastly, the EAAA intervention did not significantly affect patients' living standards and ownership of household assets. From a total of 42 possible owned assets and housing quality indicators, counts of assets were very similar in both groups. Participants in the EAAA treatment group reported on

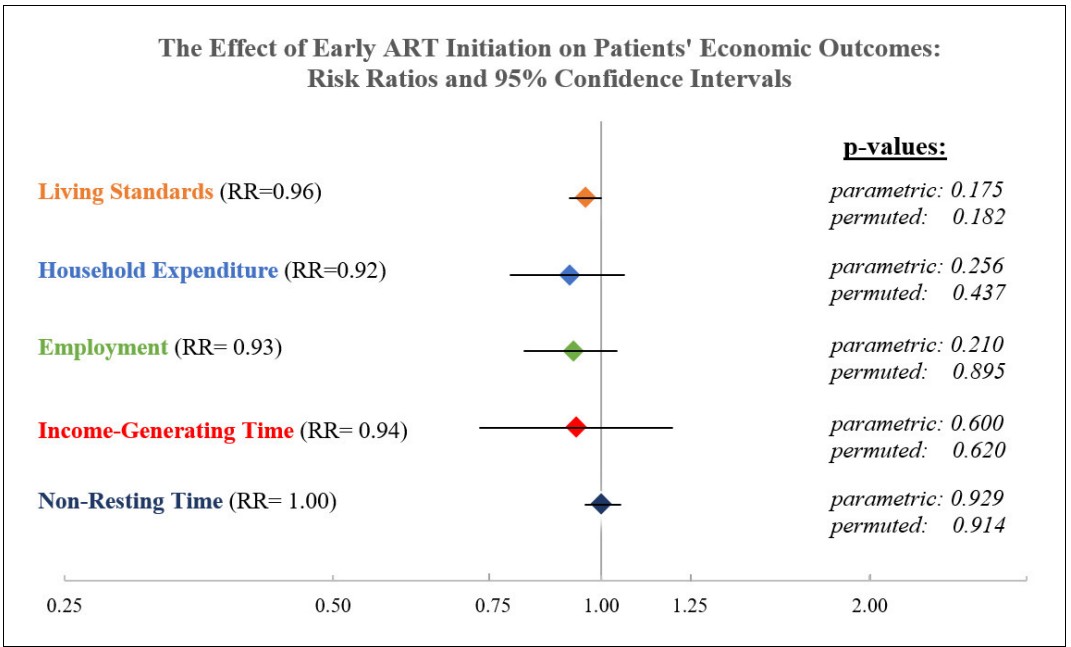

**Figure 2.** The causal effect of early ART initiation on economic outcomes. Notes: Relative Risk presented for negative binomial mixed-effect regression with random intercept by healthcare facility (cluster) and a fixed effect for study period (*Hussey and Hughes, 2007*). All models control for respondent sex, age, marital status, and highest grade completed and were grand-mean centered. Parametric p-value obtained directly from the regression output; non-parametric p-value obtained from a permutation test with 1000 replications.

The online version of this article includes the following figure supplement(s) for figure 2:

**Figure supplement 1.** Histogram: non-resting time use.
**Figure supplement 2.** Histogram: income-generating time use.
**Figure supplement 3.** Histogram: household expenditures.
**Figure supplement 4.** Household assets/living standards.
**Figure supplement 5.** Heterogeneity plots for non-resting time use.
**Figure supplement 6.** Heterogeneity plots for income-generating time.
**Figure supplement 7.** Heterogeneity plots for employment.
**Figure supplement 8.** Heterogeneity plots for household expenditures.
**Figure supplement 9.** Heterogeneity plots for household assets.

average 0.71 indicators less than participants in the standard of care group (RR = 0.96, 95% CI 0.92, 1.00, p=0.253). In alternative regression models (*Supplementary file 1F*), linear regressions (*Supplementary file 1G*) and using an alternative principal-component-weighted outcome index (see *Supplementary file 1H*), we found similar null effects.

## Heterogeneity in treatment effects

Overall, causal random forests did not identify subgroups with effects that diverged significantly from the average treatment effect. Across outcomes, most heterogeneity was found along the variables (i) patients' time on ART, (ii) number of months passed since patients' HIV diagnosis, (iii) years of education completed, and (iv) age, whereas the importance metric for patients' sex was very small, possibly due to an over-representation of women in our sample. The plots presented in *Figure 2—figure supplement 5–9* depict heterogeneity in treatment effects along these four moderating variables. It appears that the program's effect on most economic outcomes was slightly higher for patients with shorter rather than longer time on ART. However, it has to be cautioned that heterogeneity was not statistically significant for any of the four economic outcomes.

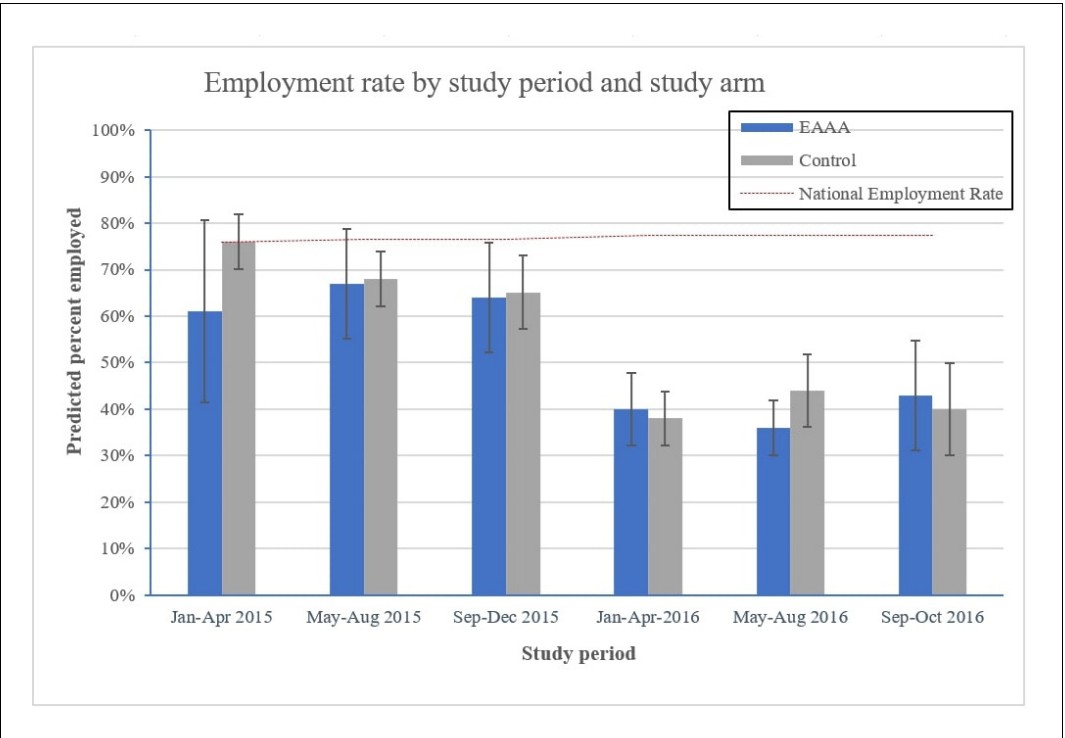

**Figure 3.** Average adjusted predictions of employment rates by period and study arm. Notes: Percent employed are the average adjusted predictions based on a logistic regression model with a time period fixed effect and a clinic-level random effect, interacting study period with trial arm, and controlling for patients' age, sex, marital status, level of education, sex (binary), marital status (binary), and their level of education (continuous, specifying the highest grade completed). Period 0 and 7 are not shown because all participants interviewed in period 0 were part of the control phase and all participants interviewed in the last period were exposed to the intervention. The national total labour force participation rate is based on World Bank data and captures the proportion of the population of working age that is economically active during the reference period of 1 year.

## Discussion

We present the first causal evaluation of the effect of immediate ART for all HIV patients on wider economic outcomes. Based on our primary results and several robustness checks, we are able to conclude that large harmful effects are very unlikely. More specifically, we found that neither patients' time use nor their employment status and living standards were positively or negatively affected by the EAAA intervention. Although we found a reduction in monthly household expenditures among patients in the EAAA group, the magnitude was small in size (−126.17 SZL, corresponding to 3% of the average monthly household expenditures in Eswatini) (*The World Bank, 2019*) and not statistically significant. Lastly, in machine-learning-supported heterogeneity analyses, we also did not find any patient subgroup for which the EAAA intervention either significantly improved or deteriorated individual- and household-level economic outcomes. Two previous publications based on the same trial have assessed how early ART initiation affected patients' health, revealing a 6% higher retention in care rate in the EAAA group but no significant differences with regard to all-cause, disease-related, and HIV-related mortality between the EAAA and the standard of care group (*Chao et al., 2020*; *Perriat et al., 2018*). While we were unable to link responses from this survey to patients' clinical data, we may still infer that more substantial health impacts would have been necessary to significantly affect patients' economic welfare. It is also possible that both the health and economic benefits of early ART initiation only materialise after a longer follow-up time, beyond the 36 months observation period covered in this trial (*May et al., 2016*). A potential alternative explanation for the absence of strong and beneficial treatment effects could relate to the broader socioeconomic conditions of the study region. Hence, if income generation opportunities are generally constrained due to given economic circumstances, people living with HIV may be

unable to find work, irrespective of whether they are healthy or not. If patients' health status does not substantially impact their earning potential, other welfare indicators such as household expenditures and living standards are also unlikely to change. However, we partly alleviate this problem by adopting a broad definition of employment by including informal and short-term work and should therefore be able to capture even small changes in participants' income generation activities.

In contrast to several prior studies (*Beard et al., 2009*; *Thirumurthy et al., 2008*; *Habyarimana et al., 2010*; *Baranov and Kohler, 2018*), our findings did not exhibit any substantial detrimental financial and economic consequences of ART initiation. At the very least, our results suggest that ART-related adverse events were not substantial enough to provoke significant drops in patients' productivity levels (*Nansseu and Bigna, 2017*). We therefore add important new empirical evidence from the perspective of patients' economic wellbeing, which – given that EAAA does not appear to have large detrimental effects on patients' economic outcomes – support the 2015 WHO recommendation to offer immediate ART initiation to all HIV-patients.

Our study has several key strengths. First, we examined a comprehensive set of outcome variables and are thus able to gain nuanced insights into participants' overall economic situation. Although the different outcomes are likely correlated, time use and employment are patient-level variables, whereas expenditures and living standards are captured at the household level. The latter two variables could thus be differently affected by the EAAA intervention depending on whether the patient is the household's main breadwinner or not. Household savings could have been another possible welfare-related aspect to assess. However, the general savings rate in Eswatini is low (*World Bank Group, 2017*) and savings are often mainly used to smooth consumption, and thus likely highly correlated with overall household expenditures. Second, we have collected very detailed information on patients' time use. Time use is a measure that is presumably highly sensitive to potential short-term changes in economic productivity and, given that we asked about the previous 24 hr, less prone to measurement error or bias from a long recall period. In view of the precise null effects for the time use outcome, we can more confidently conclude that immediate ART initiation had no harmful effects on patients' overall productivity levels. Third, and arguably most importantly, this is the first randomised study - and thus the first study to allow for causal inference under no untestable assumptions – of the impact of immediate ART initiation on indicators of patients' economic outcomes.

This study has six main limitations. First, biological data on patients' CD4 count levels and viral loads were not collected. It was thus not possible to assess whether the effects of EAAA on patients' economic outcomes were different among those patients who had a CD4 count close to the treatment threshold at the time of ART initiation. Second, participant recruitment was implemented within healthcare facilities and it is therefore possible that patients who generally attend healthcare services more regularly and reliably were overrepresented in the study sample. Third, participants were not followed-up on longitudinally, which implies that for each individual, we either had a measurement of the pre- or the post-intervention phase (but never for both). Our effect estimates are based on the comparison of patients in the standard of care phase with patients in the EAAA phase, and would turn invalid if there was significant imbalance in baseline characteristics between these two groups. However, this is unlikely in view of the sufficiently large sample size and the random selection of interview dates for each facility. Fourth, data on household expenditures and household living standards were only collected from a random subsample of 50% of patients. Given the wide confidence interval of the effect estimates for household expenditures , it is possible that we would have been able to find a significant effect for this outcome of a size that would still be meaningful to health policy makers if we had had a larger sample size. Fifth, data was based on patients' self-report. Especially with regard to household expenditures and time use, this limitation is likely to have led to some degree of measurement error due to recall problems (*Filmer and Pritchett, 2001*; *Sahn and Stifel, 2000*). In addition, while monthly expenses were summarised into 20 distinct expenditure categories to reduce interview length and cost, this may have led to further measurement imprecisions, for instance through adding up expenses for numerous individual food items into an overall category of 'total shopping for food and groceries'. Yet, we expect that these measurement errors and reporting biases occurred – on average – to an equal degree in the EAAA and standard of care group and are therefore unlikely to systematically bias our point estimates of the causal intervention effect. Lastly, the employment rate in our study sample diverged from the national employment rate during the same period. This discrepancy could be explained by (i) the

composition of our sample, which consisted of 75% female patients and is therefore not representative for the population as a whole, (ii) the temporal disaggregation into tertials, which might reflect some seasonal fluctuations in our data, and (iii) the lack of regional labour force data for the general population in the Hhohho region, rather than the aggregated national data that we have used as a reference.

This study provides the first causal evidence on the effect of immediate ART initiation on individual- and household-level economic outcomes. EAAA is unlikely to have detectable, harmful economic repercussions for HIV patients in Eswatini. This is an important finding for health policy making in that it buttresses the WHO recommendation to discard eligibility thresholds for ART from the perspective of patients' economic wellbeing – a perspective that is often ignored in the setting of clinical recommendations, yet important to those who are directly affected by these recommendations (*Govindasamy et al., 2012*; *Fox et al., 2010*).

## Materials and methods

The Maximising ART for Better Health and Zero New HIV Infections (MaxART) trial (*Walsh et al., 2017*) (NCT02909218) and the economic outcome analysis presented in here (NCT03789448) were pre-registered on ClinicalTrials.gov.

### Study setting

The study was implemented in North-Western Eswatini (formerly 'Swaziland'). 27.0% of the general population in Eswatini are HIV-positive; the highest HIV prevalence worldwide (*Government of the Kingdom of Eswatini, 2017*). The trial enrolled 14 government-managed health facilities located in the Hhohho region (see *Figure 4*). At the study's outset in 2014, all health facilities provided comprehensive HIV care and treatment according to the national adult HIV treatment guidelines effective at the time, thus initiating ART according to prescribed CD4 count levels. According to the *Annual HIV Program Report* of 2014 (*Kingdom of Swaziland, Ministry of Health, 2014*), almost 60% of HIV-patients in the Hhohho region had been initiated on ART in the year prior to the trial roll-out.

### Stepped-wedge randomised trial design

Health facilities were allocated non-randomly into seven pairs based on their geographic proximity to avoid possible contamination and based on their facility catchment size to ensure that group sizes were roughly equal. Over the course of 3 years, each of the seven pairs was randomly assigned to one of seven sequences, which determined the point in time at which each facility shifted from the standard of care (control condition) to the Early Access to ART for All (EAAA) intervention (treatment condition) (see *Table 2*). Hence, in the first period, all facilities adhered to the national standard of care while in the last period, all facilities had adopted EAAA. The randomisation was carried out by the trial statisticians. No stratification was used. This was an open-label trial in which healthcare providers and patients were unblinded to the intervention itself. However, the timing of the transition was only revealed to healthcare providers six to four weeks prior to the start of EAAA implementation.

### Control phase: standard of care

In the standard of care phase, following national treatment guidelines effective at the time, ART eligibility was restricted to patients with CD4-cell counts of <350 cells/mm$^3$ in the first 1.5 years of the study. In October 2016, the eligibility threshold was raised to CD4-cell counts < 500 cells/mm$^3$. Eligible patients were typically initiated on Eswatini's first-line ART regimen (Tenofovir (TDF) + Lamivudine (3TC) + Efavirenz (EFV)). Those with contraindications to this regimen were initiated on alternative regimens, including TDF + 3TC + Nevirapine (NVP) or Zidovudine (AZT) + 3TC + NVP (when EFV could not be used); Abacavir (ABC) + 3TC + EFV or AZT + 3TC + EFV (when TDF could not be used); ABC + 3TC + EFV or Stavudine (D4T) + 3TC + EFV (when AZT could not be used). Patients attended one private and one group counselling session prior to initiation. While same-day ART initiation was allowed according to the national Integrated HIV Management Guidelines (*Ministry of Health, Kingdom of Swaziland, 2015*), HIV diagnosis and ART initiation in the respective facilities were typically a few days apart.

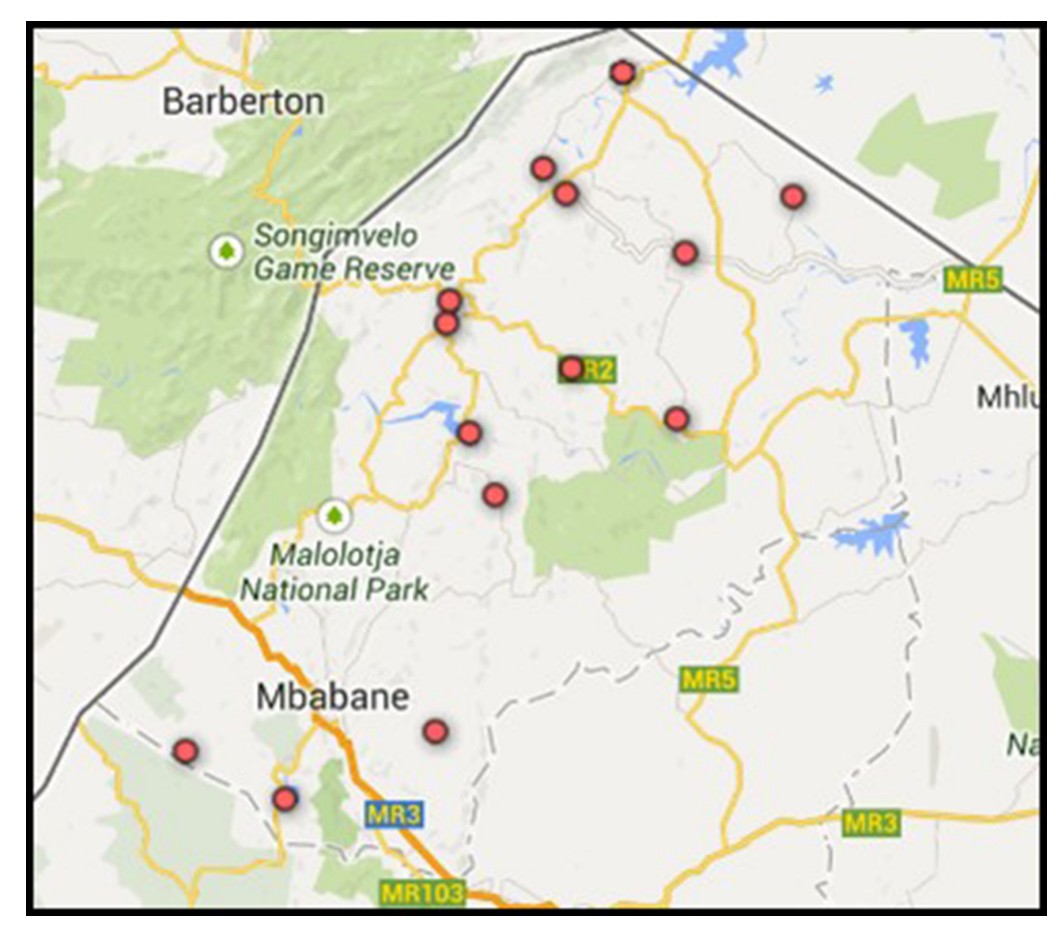

**Figure 4.** Map of the healthcare facilities that participated in the study.

## Intervention phase: early access to ART for all (EAAA)

During the EAAA intervention phase, all patients who tested HIV-positive as well as patients enrolled in pre-ART care were offered immediate ART initiation, independent of their CD4-cell count. They received one counselling session and ART initiation on the same day and further monthly counselling after initiation. As in the standard of care, patients in the EAAA programme were initiated on Eswatini's first-line treatment regimen or, if contraindicated, on the same alternative regimens detailed above.

## Data collection

Data were collected via standardised paper-based questionnaires over eight time periods (baseline and seven transitions). In every period, a sample of all HIV-care patients in each of the enrolled healthcare facilities was randomly selected. Eligibility was constrained to patients who were HIV-positive and over the age of 18 years, and who were neither pregnant nor breastfeeding. Patients were eligible irrespective of whether ART initiation could take place on the same day of HIV diagnosis or a few days thereafter. For each facility, the study team randomly selected data collection days. On these days, the study team adopted the sampling strategy of selecting the next patient entering the consultation room. This strategy yields a representative sample if the sample size is sufficiently large and the order with which patients are seen by a clinician is random. Monte Carlo simulations have shown that this sampling strategy also tends to be more efficient and unbiased compared to simple and systematic random sampling, and does not underrepresent potentially healthier patients with shorter consultations as is the case when sampling those *exiting* the consultation room (*Geldsetzer et al., 2018*). Respondents gave verbal and written consent before completing the

**Table 2.** Stepped-wedge trial design used in this study.

| Healthcare facility | Sep - Dec 2014 | Jan - Apr 2015 | May - Aug 2015 | Sep - Dec 2015 | Jan - Apr 2016 | May - Aug 2016 | Sep - Oct 2016 | Oct 2016 - Aug 2017 |
|---|---|---|---|---|---|---|---|---|
| Mshingishingini Nazarene | CONT | INT | INT | INT | INT | INT | INT | INT |
| Ntfonjeni | CONT | INT | INT | INT | INT | INT | INT | INT |
| Bulandzeni | CONT | CONT | INT | INT | INT | INT | INT | INT |
| Ndzingeni | CONT | CONT | INT | INT | INT | INT | INT | INT |
| Maguga | CONT | CONT | CONT | INT | INT | INT | INT | INT |
| Malandzela | CONT | CONT | CONT | INT | INT | INT | INT | INT |
| Pigg's Peak Hospital | CONT | CONT | CONT | CONT | INT | INT | INT | INT |
| Peak Nazarene | CONT | CONT | CONT | CONT | INT | INT | INT | INT |
| Herefords | CONT | CONT | CONT | CONT | CONT | INT | INT | INT |
| Ndvwabangeni Nazarene | CONT | CONT | CONT | CONT | CONT | INT | INT | INT |
| Sigangeni | CONT | CONT | CONT | CONT | CONT | CONT | INT | INT |
| Siphocosini | CONT | CONT | CONT | CONT | CONT | CONT | INTNT | INT |
| Horo | CONT | CONT | CONT | CONT | CONT | CONT | CONT | INT |
| Hhukwini | CONT | CONT | CONT | CONT | CONT | CONT | CONT | INT |

CONT indicates the control group phase and INT the treatment phase.

interview and were informed about their right to decline or withdraw their participation at any point in time. No prior sample size calculations were performed.

## Outcomes

We assessed the impact of the EAAA intervention on four economic outcomes. First, patients' time use during the day prior to the interview was measured by collecting detailed information on hourly activities for a cycle of 24 hr. For our analysis, we specified two outcomes that are indicative of patients' productivity levels: (i) 'non-resting time' to capture the total hours spent on activities other than sleeping and resting, and (ii) 'income-generating time' to capture the total hours spent on any income generation activities, which comprised formal employment, primary production activities in the informal sector, subsistence farming, and income generated from own businesses (i.e. from the sale of goods). The second outcome was patients' current labour market participation, categorised as 'employed' if patients were working or engaged in subsistence farming (either part- or full-time), and categorised as 'not employed' if patients were unemployed, retired or taking sick or other leave. The third outcome was patients' total past-month household expenditures on food- and non-food items, which was measured by asking each participant how much their household spends on 20 common expenditure items in a normal month (or, if the respondent preferred, in the past year) as well as on 'other usual expenses' and 'large purchases or expenses in the last 12 months' that were not mentioned in the list of common expenditure items. We opted for expenditure rather than income data because it is less affected by possible seasonal fluctuations in earnings and therefore better reflects a welfare level that households can maintain through consumption smoothing and informal borrowing (*Sahn and Stifel, 2003*; *Filmer and Pritchett, 2001*). The last outcome was household living standards, measured as an additive index counting the total number of realised housing quality indicators (12 items, e.g., drinking water inside the house, concrete walls, flush toilet, etc.) and assets owned (30 items, e.g., refrigerator, phone, TV, animals, etc.). In line with economic literature (*Sahn and Stifel, 2003*; *Filmer and Pritchett, 2001*), we also computed a principal-component-weighted index from the answers to these housing quality indicators and owned assets as an alternative metric to the additive index, reported in *Supplementary file 1H*. Information on time use and employment was captured for the full sample. In order to reduce the length of the survey, questions on household expenditures and household living standards were asked to every second participant who was interviewed.

## Data analysis

We estimated the intent-to-treat effect (ITT) by comparing patients interviewed in the standard of care phase to patients interviewed in the EAAA phase (see EXHIBIT 2). We used mixed-effects negative binomial regressions (showing the resulting risk ratios) to account for the skewed distribution of some outcome variables (income-generating time and household expenditures). For normally distributed outcome variables (non-resting time and living standards), we additionally provide results from mixed-effects linear regressions in supplementary tables. For the binary employment outcome, we also estimated risk ratios for ease of interpretation by utilising a modified poisson regression model with a robust error structure.

Following the conventional Hussey and Hughes approach, regression models included a binary indicator ('fixed effect') for each time period and a clinic-level random effect to account for clustering by clinic (*Hussey and Hughes, 2007*). While clinic-level random effects help to partly adjust for varying cluster size by assigning higher weights to larger clusters, we additionally included a permutation test to project more conservative p-values that correct for (i) the varying cluster sizes, (ii) the relatively small number of clusters, and (iii) potential violations in asymptotic properties of the regression models (*Athey and Imbens, 2016*). Specifically, for each of the main outcome models (Hussey and Hughes model with control variables), we used a permutation test (implemented in the 'swpermute' package in Stata *Thompson, 2019*) with 1000 repetitions to test for the statistical significance of the treatment effect point estimates. In supplemental results, we present a second, more flexible, model that allows for potentially heterogeneous time trends across healthcare facilities by including a random slope for time period (*Thompson et al., 2017*). Each of the two models was estimated without and with control variables, consisting of patients' age (continuous), sex (binary), marital status (binary), and their level of education (continuous, specifying the highest grade completed). While adjustment for these variables is not needed to obtain unbiased effect estimates, their inclusion in the regressions might correct for small sample biases and improve precision.

180 participants (12% of the complete random subsample) did not respond to the household expenditure questions. For this outcome, we therefore ran two regression specifications, one based on the incomplete sample (i.e. a complete case analysis) and one based on a complete sample after imputing missing observations using multivariate imputation by chained equations (MICE) (*Azur et al., 2011*). The imputation model was implemented using the '*mice*' package in Stata (*Royston and White, 2011*). We implemented the imputation with 1000 repetitions and included all variables used in the main outcome analysis as well as additional 'auxiliary' variables, which were current employment, education level, household living standards, and patients' sociodemographic characteristics. Assuming that the likelihood of a missing value is only a function of observed characteristics, the MICE procedure iteratively estimates missing values based on Markov Chain Monte Carlo techniques. It creates 1000 complete datasets to estimate missing values, which are then averaged across all datasets (*Yu et al., 2007*; *Azur et al., 2011*).

Lastly, we estimated for each of the five outcomes whether there was heterogeneity in treatment effects between different groups of patients. For this purpose, we utilised a machine learning approach in the form of a non-parametric causal forest algorithm (*Athey et al., 2019*; *Athey and Wager, 2019*; *Wager and Athey, 2018*). This approach has advantages over other subgroup tests (*Lee, 2009*; *Crump et al., 2008*) in that it (i) does not require an a priori hypothesis on the potential differential effects, (ii) increases statistical power, (iii) and yields treatment effect estimates that are asymptotically normal (*Athey and Wager, 2019*; *Wager and Athey, 2018*). In this analysis, we first assessed whether treatment effects for any subgroup were significantly different from the average treatment effect. In a second step, we explored the nature of potential heterogeneity through ordering moderating variables by their importance.

The random forest heterogeneity analysis was implemented in R 3.6.2. All other analyses were conducted in Stata 15.

# Additional information

## Funding

| Funder | Grant reference number | Author |
|---|---|---|
| Dutch Postcode Lottery in the Netherlands | | Till Bärnighausen |
| Alexander von Humboldt-Stiftung | | Jan-Walter De Neve<br>Till Bärnighausen |
| The Embassy of the Kingdom of the Netherlands in South Africa/Mozambique | | Till Bärnighausen |
| British Columbia Centre of Excellence in Canada | | Till Bärnighausen |
| Doctors Without Borders | | Till Bärnighausen |
| National Center for Advancing Translational Sciences | Award Number KL2TR003143 | Pascal Geldsetzer |
| Joachim Herz Foundation | | Janina Isabel Steinert |

The funders had no role in study design, data collection and interpretation, or the decision to submit the work for publication.

## Author contributions

Janina Isabel Steinert, Conceptualization, Data curation, Formal analysis, Validation, Investigation, Visualization, Methodology, Writing - original draft, Writing - review and editing; Shaukat Khan, Khudzie Mlambo, Fiona J Walsh, Emma Mafara, Charlotte Lejeune, Cebele Wong, Anita Hettema, Data curation, Project administration; Osondu Ogbuoji, Conceptualization, Formal analysis, Writing - review and editing; Sebastian Vollmer, Writing - review and editing; Jan-Walter De Neve, Conceptualization, Supervision, Methodology, Writing - review and editing; Sikhathele Mazibuko, Velephi Okello, Conceptualization, Data curation, Validation, Investigation, Project administration; Till Bärnighausen, Conceptualization, Resources, Data curation, Supervision, Funding acquisition, Validation, Investigation, Methodology; Pascal Geldsetzer, Conceptualization, Data curation, Formal analysis, Supervision, Funding acquisition, Validation, Investigation, Methodology, Writing - original draft, Writing - review and editing

## Author ORCIDs

Janina Isabel Steinert https://orcid.org/0000-0001-7120-0075
Fiona J Walsh https://orcid.org/0000-0003-2282-1005
Jan-Walter De Neve http://orcid.org/0000-0003-3090-8249
Pascal Geldsetzer https://orcid.org/0000-0002-8878-5505

## Ethics

Clinical trial registration NCT02909218, NCT03789448.
Human subjects: Ethical approval for this study was obtained from the Eswatini National Health Service Review Board in July 2014 (Reference Number: MH/599C/FWA 000 15267). Respondents gave verbal and written consent before completing the interview and were informed about their right to decline or withdraw their participation at any point in time. The study was further granted an exemption for non-human subjects research from the ethics review board of the Harvard T.H. Chan School of Public Health.

## Decision letter and Author response

Decision letter https://doi.org/10.7554/eLife.58487.sa1
Author response https://doi.org/10.7554/eLife.58487.sa2

# Additional files

## Supplementary files

- Source data 1. Datasets, dofiles, and R code for replication purposes.

- Supplementary file 1. Alternative regression specifications. (A) The causal effect of EAAA on non-resting time. (B) The causal effect of EAAA on income-generating time. (C) The causal effect of EAAA on employment. (D) The causal effect of EAAA on household expenditures (non-imputed sample). (E) The causal effect of EAAA on household expenditures: Imputed sample. (F) The causal effect of EAAA on asset and living standard index. (G) OLS Specifications. (H) The causal effect of EAAA on a principal component weighted asset and living standard index.

- Transparent reporting form
- Reporting standard 1. CONSORT stepped-wedge trial checklist.

## Data availability

All data generated or analysed during this study are included in the manuscript and supporting files. Source data files have been provided for Figures 2-4 and all supplementary Figures (Figures S1-S9).

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
