## [Decision Letter]

**Acceptance summary:**

This article adds an important piece to existing HIV literature by demonstrating that early ART is not associated with negative impacts on economic welfare in resource constrained settings.

**Decision letter after peer review:**

Thank you for submitting your article "A Stepped-Wedge Randomized Trial on the Impact of Early ART Initiation on HIV Patients' Economic Welfare in Eswatini" for consideration by *eLife*. Your article has been reviewed by three peer reviewers, including Joshua T Schiffer as the Reviewing Editor and Reviewer #1, and the evaluation has been overseen by a Senior Editor. The following individual involved in review of your submission has agreed to reveal their identity: Lisa Hirschhorn (Reviewer #2).

The reviewers have discussed the reviews with one another and the Reviewing Editor has drafted this decision to help you prepare a revised submission.

Summary:

Using a stepped wedge clinical trial approach in 14 clinic sites across Eswatini, this study provides the first causal evidence to show that early ART initiation does not negatively affect the economic welfare of HIV patients in a low-income country.

Overall, the reviewers agreed that this is topically important, clearly and carefully presented, and methodologically sound study. The paper assesses a critically important question, whether immediate ART impacts the economic livelihoods of treated persons, relative to the ART initiation at a CD4 T cell threshold. The study is conducted in an appropriate setting where HIV prevalence is high among adults and where a majority of the population is involved in subsistence agriculture, and economic inequality is high. The stepped wedge design with clustering by clinic site is appropriate for the scientific question, and it is great results are considered at the group and individual level.

We have no major concerns, but do have several suggestions to enhance scientific impact and clarity.

1) One major, perhaps not addressable limitation is that health outcomes were not jointly assessed in the study. While the benefits of early ART are irrefutable, it would have been helpful to show that study participants were able to remain complaint with their medicines, particularly because economic insecurity is a major driver of medical noncompliance with HIV medicines. Even if this information were available for even a subset of the study population, or in a historical local cohort, this would strengthen the impact of the work.

2) Data collection:

a) The following phrase is confusing: "individuals were randomly selected". Were these individuals then followed longitudinally or was the selection performed independently at each visit? If the latter strategy was performed, the authors should dedicate a paragraph in the Discussion towards explaining this limitation.

b) The description of the sampling of individuals to complete surveys for the data collection is contradictory. On the one hand it is stated that a random sample of participants was selected at each time point, but on the other hand it is said that an interviewer visited a clinic on a randomly selected day and then interviewed consecutive patients. This apparent contradiction should be resolved. It should be clarified whether any individuals were interviewed more than once.

c) Please provide more detail of the how the 7 pairs of clusters were formed. It seems this was done in an essentially deterministic way, at least definitely not randomly. In this sense then the trial really had only 7 “clusters” when considering the meaning of a cluster for analysis. Were the analysis approaches all conducted as if 14 clusters had been randomly allocated to 7 sequences, including the permutations tests? This should be noted in the Materials and methods and mentioned as a limitation in the Discussion.

d) When describing the trial in the Materials and methods the authors should add a description of the nature of how individuals are exposed to the trial. Is the appropriate phrase to describe this as an open cohort trial?

3) The post-hoc power calculations should be removed as these are illogical, and a statement added to the Materials and methods that no sample size calculation was performed. All mention of power should be removed from the article.

4) While the diversity of selected economic endpoints is appreciated, why not measure change in income and/or savings? These seem like the most direct assessments of economic well-being. Please explain in the Discussion.

5) Please provide clarification on the enrolment sites. Prior to the intervention study, was same day ART initiation established and commonly practiced? Understanding the percent of clinic attendees on ART from the clinics before they initiated the study would help understand the percent of populations who transitioned from pre-ART to ART at study initiation.

6) In regard to enrolment, are only those who actually started on ART included from the sites when they started same day or is it all people who were approached about the study?

7) It would be helpful if the authors could explore the impact of the intervention based on duration of ART as some of the sites would be able to contribute data from individuals on ART for considerably longer, which may allow for some exploration of the economic effect (or absence) over time. This is noted in discussion, but for some patients, time on ART at the point of survey would have been considerably longer.

8) Data presentation:

a) Figure 2 is not very useful. Because there are only 2 facilities per cluster, the variance is not a useful metric. I suggest listing the cluster size of each facility in the figure. I would suggest having some method to call out clusters with small size so the reader can get a better sense of proportion of sites that were under-enrolled. The font size is also way too small.

b) Table 3 should be a figure with RR and 95% CI projected. It would be much easier for the reader to process the data if presented this way.

c) For Figure 4, is this decline in employment consistent with regional economic trends? If so, this should be included in the figure (perhaps as a line showing % unemployment at the national level). Of not, then this discrepancy should be explained.

d) All supplementary figures are poorly labelled and hard to read.

9) While the results seem robust, sensitivity analyses should be performed to assess the degree of misclassification due to patient self-report bias that could nullify the study's results. Sensitivity analyses should also be considered to assess whether variance in cluster size could result in misleading results.

10) Discussion:

a) The language in the first paragraph of the Discussion is poor and should be improved. Phrases such as "precise null effects" and "confident in our null result" should be removed, as these are unclear. Please remove any mention of statistical power. The authors should describe how they found little impact of the intervention on economic outcomes and focus on the bounds of the confidence intervals.

b) The authors should reflect on how much of the no impact observed was related to high unemployment, which may have limited any negative impacts on ability to work?

c) It should be mentioned as a limitation that data were collected only from those attending clinics and that the sampling method would oversample regular attenders in comparison to those who attend more erratically.

---

## [Author Response]

We have no major concerns, but do have several suggestions to enhance scientific impact and clarity.1) One major, perhaps not addressable limitation is that health outcomes were not jointly assessed in the study. While the benefits of early ART are irrefutable, it would have been helpful to show that study participants were able to remain complaint with their medicines, particularly because economic insecurity is a major driver of medical noncompliance with HIV medicines. Even if this information were available for even a subset of the study population, or in a historical local cohort, this would strengthen the impact of the work.

Thank you very much for highlighting this. In response, we reference two papers that were published on the same trial, assessing the impact of early ART initiation on (i) patients’ retention in ART care and (ii) patient mortality.

We have now added a paragraph in our Discussion section, summarising the wider trial results and putting them in context with regards to the economic outcomes reported in here:

“Two previous publications based on the same trial have assessed how early ART initiation affected patients’ health, revealing a 6% higher retention in care rate in the EAAA group but no significant differences with regards to all-cause, disease-related, and HIV-related mortality between the EAAA and the standard of care group. While we were unable to link responses from this survey to patients’ clinical data, we may still infer that more substantial health impacts would have been necessary to significantly affect patients’ economic welfare. It is also possible that both the health and economic benefits of early ART initiation only materialise after a longer follow-up time, beyond the 36-months observation period covered in this trial.”

2) Data collection:a) The following phrase is confusing: "individuals were randomly selected". Were these individuals then followed longitudinally or was the selection performed independently at each visit? If the latter strategy was performed, the authors should dedicate a paragraph in the Discussion towards explaining this limitation.

Thank you. To clarify: individual participants were not followed longitudinally. Rather, we randomly selected a data collection date for each facility, which then determined whether patients were interviewed during their standard of care phase or during the EAAA intervention phase (see: **“**For each facility, the study team randomly selected data collection days. On these days, the study team adopted the sampling strategy of selecting the next patient entering the consultation room”).

As suggested, we have now added a paragraph on this in our Discussion section:

“Third, participants were not followed-up on longitudinally, which implies that for each individual, we either have a measurement of the pre- or the post-intervention phase (but never for both). Our effect estimates are based on the comparison of patients in the standard of care phase with patients in the EAAA phase, and would turn invalid if there was significant imbalance in baseline characteristics between these two groups. However, this is unlikely in view of the sufficiently large sample size and the random selection of interview dates for each facility.”

b) The description of the sampling of individuals to complete surveys for the data collection is contradictory. On the one hand it is stated that a random sample of participants was selected at each time point, but on the other hand it is said that an interviewer visited a clinic on a randomly selected day and then interviewed consecutive patients. This apparent contradiction should be resolved. It should be clarified whether any individuals were interviewed more than once.

Thank you and we acknowledge that we had not explained our sampling procedure in sufficient detail. Apologies for this. Our strategy of sampling the next patient entering the consultation room is not a “simple random sample” but still results in a representative sample with a sufficiently large sample (which we are confident is given here). Based on Monte Carlo simulations and varying assumptions with regards to the number of consultation rooms, interviewers, and consultation length, sampling the next patient entering the consultation room was shown to be the most efficient and unbiased sampling method relative to simple and systematic random sampling, as well as compared to sampling the next patient *exiting* the consultation room (see Geldsetzer et al., 2018).

To explain this in the manuscript, we have now added the following paragraph:

“For each facility, the study team randomly selected data collection days. On these days, the study team adopted the sampling strategy of selecting the next patient entering the consultation room. This strategy yields a representative sample if the sample size is sufficiently large and the order with which patients are seen by a clinician is random. Monte Carlo simulations have shown that this sampling strategy also tends to be more efficient and unbiased compared to simple and systematic random sampling, and does not underrepresent potentially healthier patients with shorter consultations as is the case when sampling those exiting the consultation room.”

c) Please provide more detail of the how the 7 pairs of clusters were formed. It seems this was done in an essentially deterministic way, at least definitely not randomly. In this sense then the trial really had only 7 “clusters” when considering the meaning of a cluster for analysis. Were the analysis approaches all conducted as if 14 clusters had been randomly allocated to 7 sequences, including the permutations tests? This should be noted in the Materials and methods and mentioned as a limitation in the discussion.

Thank you. While the timing for the transition from standard of care to EAAA was randomised, the selection of facilities and their matching was non-random. The matching of facilities was done to ensure that the catchment sizes were relatively equal and that the paired facilities were close to each other as an attempt to avoid contamination. We have therefore added the following explanation to our Materials and methods section:

“Health facilities were allocated non-randomly into seven pairs based on their geographic proximity to avoid possible contamination and based on the facility catchment size to ensure that group sizes were roughly equal.”

With regards to your second question: all analyses were conducted with fourteen rather than with seven clusters. The rationale for this was that we assumed outcomes to be correlated within individual facilities rather than within facility pairs. Given that facilities were matched based on logistical considerations, they are likely to differ in terms of other characteristics, including their catchment size, healthcare staff characteristics, and facility infrastructure and equipment. Despite this reasoning, we have now also conducted the main outcome analyses by clustering for the unit of analysis, i.e. the facility pairs. The results are presented in Author response table 1 and indicate that there are only very minor differences in effect estimates (specifically in the size of the confidence interval) between the “14 cluster model” (14 CL) and the “7 cluster model” (7 CL).

**Author response table 1:** Comparisons between analyses with 14 vs. 7 clusters

d) When describing the trial in the methods the authors should add a description of the nature of how individuals are exposed to the trial. Is the appropriate phrase to describe this as an open cohort trial?

Thanks, this is now specified in more detail:

“This was an open-label trial in which healthcare providers and patients were unblinded to the intervention itself. However, the timing of the transition was only revealed to healthcare providers six to four weeks prior to the start of EAAA implementation.”

3) The post-hoc power calculations should be removed as these are illogical, and a statement added to the Materials and methods that no sample size calculation was performed. All mention of power should be removed from the article.

Thank you for noting this. We have now deleted any reference to post-hoc power calculations in the manuscript and have also removed these from the supplements. As suggested, we have added the following statement to our Materials and methods section: “No prior sample size calculations were performed.”

4) While the diversity of selected economic endpoints is appreciated, why not measure change in income and/or savings? These seem like the most direct assessments of economic well-being. Please explain in the Discussion.

Thank you and this is indeed important. In response, we have added a justification of why we consider expenditure data to be superior to income data:

“We opted for expenditure rather than income data because it is less affected by possible seasonal fluctuations in earnings and therefore better reflects a welfare level that households can maintain though consumption smoothing and informal borrowing.”

With regards to savings, we have added the following explanation to our Discussion section:

“Household savings could have been another possible welfare-related aspect to assess. However, the general savings rate in Eswatini is low and savings mainly used to smooth consumption, and thus likely highly correlated with overall household expenditures.”

5) Please provide clarification on the enrolment sites. Prior to the intervention study, was same day ART initiation established and commonly practiced? Understanding the percent of clinic attendees on ART from the clinics before they initiated the study would help understand the percent of populations who transitioned from pre-ART to ART at study initiation.

Thanks a lot, we agree that this is important contextual information. Same-day ART initiation was generally allowed according to the National Integrated HIV Management Guidelines of 2015. According to the Ministry of Health and WHO guidelines for ART initiation, patients were required to attend at least one group counselling session and one individual counselling session prior to ART initiation. These sessions could technically happen on the same day (i.e. on the day of HIV diagnosis), however, most facilities left some days between diagnosis and ART initiation. Therefore, same-day ART initiation was not commonly practiced in the standard of care phase. After transition to “test and treat”, all patients regardless of CD4 count were offered same day ART.

We have now specified this in the manuscript:

“Patients attended one private and one group counselling session prior to initiation. While same-day ART initiation was allowed according to the national Integrated HIV Management Guidelines, HIV diagnosis and ART initiation in the respective facilities were typically a few days apart.”

Thank you also for noting that it would be important to know how many HIV-positive patients had already been initiated on ART prior to the beginning of this study. We agree that this is important contextual information. Unfortunately, we do not have detailed information on this from our specific health facilities, but instead we draw on regional data presented in Eswatini’s Annual HIV Program Report from 2014. According to this, 3,986 patients aged 0-14 years and 41,820 patients aged 15 years and older had initiated ART in the Hhoho region by 2014. With a regional population size of roughly 300,000 and an HIV prevalence rate of 27%, we can infer that almost 60% of HIV-patients were on ART prior to the trial roll-out. In light of this, we have added the following paragraph to our Materials and methods:

“According to the Annual HIV Program Report of 2014, almost 60% of HIV-patients in the Hhohho region had been initiated on ART in the year prior to the trial roll-out.”

6) In regard to enrolment, are only those who actually started on ART included from the sites when they started same day or is it all people who were approached about the study?

Same day ART initiation was not defined as an eligibility criterion. We agree that our eligibility criteria were not specified in sufficient detail and have now added the following information:

“Eligibility was constrained to patients who were HIV-positive and ART-naïve, who were over the age of 18 years, and who were neither pregnant nor breastfeeding. Patients were eligible irrespective of whether ART initiation could take place on the same day of HIV diagnosis or a few days thereafter.”

7) It would be helpful if the authors could explore the impact of the intervention based on duration of ART as some of the sites would be able to contribute data from individuals on ART for considerably longer, which may allow for some exploration of the economic effect (or absence) over time. This is noted in Discussion, but for some patients, time on ART at the point of survey would have been considerably longer.

Thanks a lot for highlighting this and we agree that this is indeed interesting to look at.

We have assessed whether heterogeneity in treatment effects varies by patients’ time on ART and have not found any differential treatment effects between patients with more or less time on ART. Our heterogeneity plots (see Figure 3—figure supplements 5-9) tentatively suggest that patients with shorter ART duration may benefit slightly more from the EAAA intervention (in terms of their economic welfare), however, we could not confirm any significant subgroup differences in our machine-learning causal random forest approach.

We have now included a more detailed discussion of this in the manuscript:

“Across outcomes, most heterogeneity was found along the variables (i) patients’ time on ART, (ii) number of months passed since patients’ HIV diagnosis, (iii) years of education completed, and (iv) age, whereas the importance metric for patients’ sex was very small, possibly due to an over-representation of women in our sample. The plots presented in Figure 3—figure supplements 5-9 depict heterogeneity in treatment effects along these four moderating variables. It appears that the program’s effect on most economic welfare outcomes was slightly higher for patients with shorter rather than longer time on ART. However, it has to be cautioned that heterogeneity was not statistically significant for any of the four economic outcomes.”

8) Data presentation:a) Figure 2 is not very useful. Because there are only 2 facilities per cluster, the variance is not a useful metric. I suggest listing the cluster size of each facility in the figure. I would suggest having some method to call out clusters with small size so the reader can get a better sense of proportion of sites that were under-enrolled. The font size is also way too small.

Thank you and we agree. We have edited Figure 2 based on your suggestions, have removed the variance, and present the cluster size for each facility separately. In addition, the font size has been increased for improved readability.

b) Table 3 should be a figure with RR and 95% CI projected. It would be much easier for the reader to process the data if presented this way.

We appreciate this really good advice. We have replaced Table 3 with Figure 3 and agree that this is now much more reader-friendly.

c) For Figure 4, is this decline in employment consistent with regional economic trends? If so, this should be included in the figure (perhaps as a line showing % unemployment at the national level). Of not, then this discrepancy should be explained.

Thank you very much for highlighting this important point. As you suggested, we have now added a line into the figure showing the national employment rate as a reference (see Figure 4, line in red). From this, it can be concluded that the employment rate observed in our study population was not in line with the national employment rate between 2015-2016. We have included a note on this in the Results section:

“We observed a decline in general employment rates over the entire study period, from 0.64 (SD=0.48) in study period 0 to 0.35 (SD=0.48) in study period 7 (see Figure 4). The employment trend observed in our study population stands in contrast to the national employment rate during the same period, which remained constant at 77-78%.”

Given the discrepancy and following your advice, we have also added this paragraph to our Discussion section:

“Lastly, the employment rate in our study sample diverged from the national employment rate during the same period. This discrepancy could be explained by (i) the composition of our sample, which consisted of 75% female patients and is therefore not representative for the population as a whole, (ii) the temporal disaggregation into tertials, which might reflect some seasonal fluctuations in our data, and (iii) the lack of regional labour force data for the general population in the Hhohho region, rather than the aggregated national data that we have used as a reference.”

d) All supplementary figures are poorly labelled and hard to read.

Thank you and apologies for this. We have now edited all supplementary figures (Figure 3—figure supplements 1-9) by increasing the size and resolution and specified more detailed titles for the x- and y-axes.

9) While the results seem robust, sensitivity analyses should be performed to assess the degree of misclassification due to patient self-report bias that could nullify the study's results. Sensitivity analyses should also be considered to assess whether variance in cluster size could result in misleading results.

Thanks for highlighting this. With regards to the variance in cluster size, we report more conservative p-value estimates that are obtained from the permutation test that explicitly adjusts for the varying cluster size. We have now highlighted this in the manuscript:

“While clinic-level random effects help to partly adjust for varying cluster size by assigning higher weights to larger clusters, we additionally included a permutation test to project more conservative p-values that correct for (i) the varying cluster sizes, (ii) the relatively small number of clusters, and (iii) potential violations in asymptotic properties of the regression models.”

In addition, we would like to emphasise that unequal cluster sizes are not a threat to accuracy (unbiasedness) and consistency as such but that they may decrease statistical power (Martin, Hemming and Girling, 2019).

With regards to possible self-report biases, we agree with you that these could add some measurement error to our outcomes, most likely in result of recall issues with regards to the household expenditure and time use measurements. However, we strongly believe that such measurement errors would occur, on average, to a similar extent in both study arms. At least it is not evident to us why participants would exhibit differential reporting biases with regards to their economic welfare depending on whether they have been exposed to the intervention (immediate ART initiation) or the standard of care (CD4-count-dependent ART initiation). Therefore, we contend that this random noise is highly unlikely to bias or invalidate our results. This is now highlighted more specifically in the manuscript:

“Fifth, data was based on patients’ self-report. Especially with regards to household expenditures and time use, this limitation is likely to have led to some degree of measurement error due to recall problems. In addition, while monthly expenses were summarised into 20 distinct expenditure categories to reduce interview length and cost, this may have led to further measurement imprecisions, for instance through adding up expenses for numerous individual food items into an overall category of “total shopping for food and groceries”. Yet, we expect that these measurement errors and reporting biases occurred, on average, to an equal degree in the EAAA and standard of care group and are therefore unlikely to systematically bias our point estimates of the causal intervention effect.”

In addition to that, our estimates are relatively precise and confidence intervals, especially for non-resting time use and household living standards, are narrow, which suggests that the results are unlikely to be driven by substantial measurement error. More importantly, our findings are consistent in that they point to null effects across the different outcome categories.

10) Discussion:a) The language in the first paragraph of the Discussion is poor and should be improved. Phrases such as "precise null effects" and "confident in our null result" should be removed, as these are unclear. Please remove any mention of statistical power. The authors should describe how they found little impact of the intervention on economic outcomes and focus on the bounds of the confidence intervals.

Thanks, point taken. Please see below the revised paragraph:

“We present the first causal evaluation of the effect of immediate ART for all HIV patients on wider economic outcomes. Based on our primary results and several robustness checks, we are able to conclude that large harmful effects are very unlikely. More specifically, we found that neither patients’ time use nor their employment status and living standard were positively or negatively affected by the EAAA intervention. Although we found a reduction in monthly household expenditures among patients in the EAAA group, the magnitude was small in size (-126.17 SZL, corresponding to 3% of the average monthly household expenditures in Eswatini) (Thompson, 2019) and not statistically significant. Lastly, in machine-learning-supported heterogeneity analyses, we also did not find any patient subgroup for which the EAAA intervention either significantly improved or deteriorated overall economic welfare.”

b) The authors should reflect on how much of the no impact observed was related to high unemployment, which may have limited any negative impacts on ability to work?

Thank you for raising this good point. We have added the following paragraph to our Discussion section:

“A potential alternative explanation for the absence of strong and beneficial treatment effects could relate to the broader socioeconomic conditions of the study region. Hence, if income generation opportunities are generally constrained due to given economic circumstances, HIV-patients may be unable to find work, irrespective of whether they are healthy or not. If patients’ health status does not substantially impact their earning potential, other welfare indicators such as household expenditures and living standards are also unlikely to change. However, we partly alleviate this problem by adopting a broad definition of employment by including informal and short-term work and should therefore be able to capture even small changes in participants’ income generation activities.”

c) It should be mentioned as a limitation that data were collected only from those attending clinics and that the sampling method would oversample regular attenders in comparison to those who attend more erratically.

Thank you, we agree. This has been added to our limitation section:

“Second, participant recruitment was implemented within healthcare facilities and it is therefore possible that patients who generally attend healthcare services more regularly and reliably are overrepresented in the study sample.”